# Application of Optical Coherence Tomography (OCT) to Analyze Membrane Fouling under Intermittent Operation

**DOI:** 10.3390/membranes13040392

**Published:** 2023-03-30

**Authors:** Song Lee, Hyeongrak Cho, Yongjun Choi, Sangho Lee

**Affiliations:** 1School of Civil and Environmental Engineering, Kookmin University, 77, Jeongneung-ro, Seongbuk-gu, Seoul 02707, Republic of Korea; 2Desalination Technologies Research Institute (DTRI), Saline Water Conversion Corporation (SWCC), WQ36+XJP, Al Jubayl 35417, Saudi Arabia

**Keywords:** membrane, fouling, intermittent operation, reverse osmosis, optical coherence tomography

## Abstract

There is increasing interest in membrane systems powered by renewable energy sources, including solar and wind, that are suitable for decentralized water supply in islands and remote regions. These membrane systems are often operated intermittently with extended shutdown periods to minimize the capacity of the energy storage devices. However, relatively little information is available on the effect of intermittent operation on membrane fouling. In this work, the fouling of pressurized membranes under intermittent operation was investigated using an approach based on optical coherence tomography (OCT), which allows non-destructive and non-invasive examination of membrane fouling. In reverse osmosis (RO), intermittently operated membranes were investigated by OCT-based characterization. Several model foulants such as NaCl and humic acids were used, as well as real seawater. The cross-sectional OCT images of the fouling were visualized as a three-dimensional volume using Image J. The OCT images were used to quantitatively measure the thickness of foulants on the membrane surfaces under different operating conditions. The results showed that intermittent operation retarded the flux decrease due to fouling compared to continuous operation. The OCT analysis showed that the foulant thickness was significantly reduced by the intermittent operation. The decrease in foulant layer thickness was found to occur when the RO process was restarted in intermittent operation.

## 1. Introduction

Water is the basis of life and a necessity for everyone. However, it is becoming an increasingly scarce and degraded natural resource for millions of people around the world. Providing a rapidly growing population with sufficient water for a variety of uses has become one of the greatest challenges of recent years [1]. Desalination of seawater and brackish water is one of the promising solutions to secure fresh water supply because it is not limited by the availability of conventional water resources [2]. Reverse osmosis (RO) is a key technology for energy-efficient and cost-effective desalination [3]. RO desalination is not only used for the supply of domestic water, but also for the supply of water for industrial and agricultural purposes [4,5].

In particular, there has been recent interest in technologies that combine renewable energy (RE) with RO desalination to provide small-scale domestic water supply [6]. This method can not only provide a stable water supply in areas where there are no existing water supply facilities, but it can also reduce the cost of electricity by using RE sources such as solar and/or wind power [6,7]. Nevertheless, RE-powered RO desalination technology has its challenges associated with the intermittency and the variability of the renewable energy sources. Due to the intermittency and variability of renewable energy, these systems require energy storage using batteries, which are expensive and have a relatively short life [7]. These issues make battery-less RE-powered RO systems more attractive [8]. However, fluctuations in the input of renewable energy sources inherently result in either intermittent or variable operation of these RO systems [7,8,9]. Such operation affects the permeate production and permeate quality, which makes it difficult to predict the RO performance [10]. Accordingly, understanding how these systems behave during intermittent operation is important to keep them stable.

A handful of works have been carried out to investigate the effect of the intermittent or variable operation on fouling in RO systems [11,12,13]. In these studies, the changes in flux or applied pressure in time were measured in lab-scale or full-scale RO processes under intermittent/variable operation modes. Nevertheless, there is still limited information available on the effect of intermittent operation on RO fouling. This is due to the limitations of existing fouling analysis techniques such as scanning electron microscopy (SEM) and atomic force microscopy (AFM). They can only be applied after the membrane process is completed. While these “autopsy” approaches are still useful, it is necessary to monitor membrane fouling in real time during the intermittent operation. A direct observation of real-time RO fouling has been attempted, which could only provide images on the surface of the foulants [14]. Due to lack of in-depth real-time analysis methods, it is difficult to figure out how fouling is progressing and what factors affect the fouling propensity in real time.

Accordingly, optical coherence tomography (OCT) has recently received increased attention as a novel approach for real-time analysis of membrane fouling [15]. OCT is a quick, sensitive, non-invasive, user-friendly device that provides high-resolution images [16]. OCT is an interferometric technique based on the interference between a split and later recombined broadband optical field [17,18,19]. OCT can measure a large area and has fast data compared to other focused optical microscopes. In OCT, a two- or three-dimensional image is obtained by performing multiple depth scans. These scans are performed as the beam is scanned laterally in either one or two orthogonal directions. Typically, a two-dimensional cross-sectional image can contain approximately 500 depth scans covering a width of 5 mm [17]. Compared to some other tomography techniques (e.g., X-ray computed tomography), OCT can perform cross-sectional imaging at a relatively high speed [20,21,22]. The most attractive feature of OCT is the ability to perform “optical slicing” at a relatively high resolution (~2 μm), providing a powerful tool to non-invasively detect the 3D structure of a semi-transparent layer [15].

Due to its advantages, OCT has applications in ophthalmology [18,23] and can be used to analyze fouling in membrane processes [24,25]. In UF membrane operation, OCT has been explored to quantitatively interpret the formation of a cake layer during a membrane process using silica nanoparticles and bentonites [19]. In NF/RO operation, membrane fouling data were acquired by OCT for deep neural network modeling [26]. OCT has also been used to quantitatively analyze the scaling of CaSO_4_ in membrane distillation (MD) [27]. It has also been used to analyze internal fouling in oil–water separation by membrane filtration [28]. Although OCT alone cannot identify the type of fouling, it may be combined with other techniques to provide information on foulant types and mechanisms.

The use of the OCT technology for membrane processes is still in its early stage. Thus, there has not been any application of OCT in full-scale RO systems. It is also challenging to apply the OCT technology to examine fouling in commercial spiral wound modules. Nevertheless, it could be overcome in several ways. For instance, OCT could be combined with the membrane fouling simulators (MFS) [29] or the “canary cell” [30], which are devices for fouling studies and show comparable results to large-scale membrane installations. It might also be possible to design a special spiral wound module with a transparent window, which allows the direct application of the OCT.

Nevertheless, it is necessary to optimize OCT methodology for target membrane systems and develop techniques for the analysis of images and the interpretation of data. This is especially important when OCT is adopted to a special system such as an intermittent RO system. In this study, an OCT method for non-destructive acquisition and analysis of membrane fouling images was developed for a laboratory-scale brackish water RO (BWRO) system. This method was then applied to investigate the membrane fouling characteristics under intermittent operation. In the experiments, synthetic feed water containing salts and organic matter was prepared to simulate the fouling during the operation. Fouling variations were observed and analyzed during stopping and restarting during intermittent operation to investigate the causes of fouling reduction. The membrane fouling layer was compared and analyzed with the flow by the stopping condition during five days of intermittent operation. AFM and 3D OCT images were collected to understand the morphological characteristics of the foulants.

## 2. Materials and Methods

### 2.1. Feed Water and RO Membrane

Sodium chloride (NaCl, Samchun, Pyeongtaek, Republic of Korea) and humic acid (Sigma Aldrich, St. Louis, MO, USA) were used as model foulants. Deionized water was produced by a water purification system (Human power I+, Human corporation, Seoul, Republic of Korea). Synthetic feed water was prepared by adding 16.0 g of NaCl and 30 mg of humic acid to 1 L of deionized (DI) water, which was intended to accelerate flux decline and membrane fouling. This may also simulate the conditions near the end of the tail elements in the BWRO process, where the TDS is 4~6 times higher than that of the feed solution. Thin-film composite RO membranes (RE4040-BE, Toray Chemical Korea, Seoul, Republic of Korea), which have a nominal salt rejection of 99.7% under standard conditions, were used for the experiments. The membrane samples were stored in deionized (DI) water at 4 °C and soaked in DI water at room temperature for 24 h before each test.

### 2.2. Experimental Set Up

The fouling behavior of the RO membrane was evaluated in a bench-scale filtration system, as shown schematically in Figure 1a.

This test system consisted of a cross-flow membrane cell with a flow channel (dimensions 60 mm × 60 mm × 3 mm). A membrane coupon was put into the membrane cell for each experiment. The effective area of the RO membrane was 36 cm^2^. As shown in Figure 1b, a transparent acrylic window was placed on the top of the membrane cell to allow direct observation of the membrane during the experiment. A high-pressure pump (Hydra cell, Minneapolis, MN, USA) was used to supply feed water to the membrane cell. The operating pressure was set to 20 bar for the experiments and the cross-flow velocity in the feed channel of the membrane cell was kept constant at 0.167 m/s. The feed water was maintained at room temperature (22 ± 1 °C) by a heat exchanger coil connected to the water bath. The permeate flow rate was monitored by a flow meter connected to a data acquisition system. A total recycling operation was carried out in which the concentrate and the permeate were sent back to the feed tank.

The RO experiments were conducted in three cases: in the first case, the membrane was operated continuously for 6 h; in the second and third cases, the membrane was shut down for 2 h after 4 h of operation to investigate changes in the membrane foulant layer and flow recovery upon resumption of operation. In the second case, the membrane was stored in the feed water during the shutdown, and in the third case, the membrane was washed with distilled water for 5 min and then stored in distilled water for the remaining shutdown time. In real processes, permeate is used to clean and store membranes. However, this lab-scale equipment does not store the permeate separately, so DI water was used instead of the permeate. In the long-term operation experiment, 8 h of operation and 16 h of shutdown were repeated for 5 days, which simulated the intermittent operation cycles for RE-powered RO systems. The operations were performed under the same conditions as in the previous cases: in the first case, the membrane was operated continuously for 40 h, in the second case, the membrane was stored in raw water during the operation stop time (16 h), and in the third case, the membrane was washed with distilled water for 5 min and then stored in distilled water for the remaining operation stop time. The operating conditions of the membrane are summarized in Table 1.

### 2.3. Optical Coherence Tomography (OCT)

An OCT system (OQ LabScope, Lumedica, Durham, NC, USA) was applied to analyze foulants on the RO membrane surface in real time during the operation. Figure 1 shows the OCT system attached to the RO test apparatus. This system has the ability to acquire 8-bit grayscale images of 512 × 512 pixels in real time. The software supplied by the manufacturer was installed and used to store and analyze the acquired image files. The actual physical size of a single pixel was 3.8 μm. The A-scan and B-scan line rates were 34 kHz and 22 s, respectively.

### 2.4. Image Analysis

The acquired images from the OCT system require several steps of image analysis, including the correction of imaging defects (e.g., non-uniform illumination, electronic noise, glare effect), enhancement of images, and segmentation of objects in the image and image measurements [31]. For the image analysis, Image J, a Java-based, freely distributed image processing program developed by the National Institutes of Health and the Laboratory for Optical and Computational Instrumentation (LOCI) at the University of Wisconsin [32], was used in this study. Image J has been used in many scientific studies and projects, mainly focused on life sciences [33,34,35], and can also be applied to membrane technology. When an image was taken using OCT, it included several layers: a membrane layer, a foulant layer, and raw water passing on the membrane. These layers were separated to extract the foulant layer using Image J, allowing the analysis of foulant properties on the membrane.

Figure 2 illustrates the procedures for the image analysis by Image J, which are based on previous works [36,37]. A raw image obtained from OCT initially has noise due to incorrect pixel intensity, as shown in Figure 2a. To deal with this, the Unsharp mask filter was used to enhance the sharpness of the image, and the result is shown in Figure 2b [37]. Next, the contrast-limited adaptive histogram equalization (CLAHE) was used to adjust the pixel intensity to equalize the histogram across the image, which is shown in Figure 2c [37,38]. Then, the Gaussian blur function was applied to remove noise in the image and eliminate outlier pixels, resulting in the image shown in Figure 2d [36]. Finally, the threshold function was used to separate the membrane fouling layer (Figure 2e) and convert it to a grayscale image to distinguish it (Figure 2f) [39].

In addition, 3D images were obtained using OCT by continuous volume scanning. A total of 30 images were used to create one 3D image by applying the 3D Volume function in Image J. The removal of the noise and the extraction of the foulant layer were also carried out. The vertical and horizontal lengths of the 3D images were set to 1946 μm and 114 μm, respectively. Only qualitative analysis was performed so these 3D images could show the overall appearance of the foulant layer on the membrane surface.

### 2.5. Atomic Force Microscopy (AFM)

After the RO experiments, the membrane surface was examined using an AFM (Park Systems NX10, Santa Clara, CA, USA). The AFM scan of the sample was performed in the contact mode, where the tip scans the sample in close contact with the surface [40]. The scan area for the samples was set to 50 μm × 50 μm.

## 3. Results and Discussion

### 3.1. Comparison of Flux in Continuous and Intermittent RO Operations

The bench-scale RO system was operated for 600 min in various modes, including continuous and intermittent operations. The feed solution containing 16.0 g/L of NaCl and 30 mg/L of humic acid was used to observe the fouling behaviors. The initial flux ranged from 22 L/m^2^-h to 23 L/m^2^-h at 20 bar, which decreased with time due to fouling. In the case of the continuous operation (Case 1), the flux decreased by 12.5% after 60 min of the operation, as shown in Figure 3a. After 240 min of operation, 120 min of shutdown in the feed water, and then restart (Case 2), the flux reduction after 600 min was approximately 10.4%, which is shown in Figure 3b. When DI water was used instead of the feed water under the same intermittent operation conditions (Case 3), less flux reduction (approximately 8.9%) was achieved, as can be seen in Figure 3c. These results indicate that the intermittent operation can retard RO fouling.

It is noteworthy that the flux was improved after the shutdown and restart of the system, which is indicated in Figure 3d. The intermittent operation using the feed water (Case 2) showed an improvement in flux of about 3.2% from before to after the restart compared to the continuous operation (Case 1). The intermittent operation with DI water (Case 3) showed a greater improvement in flux, approximately 8.6%. Accordingly, the difference in the flux reduction between Case 2 and Case 3 can be attributed to the difference in the flux improvement after the shutdown. Fouling reduction by shutting down and restarting a membrane process has also been reported in the literature [41]. The normalized flux values at 600 min of the operation for Case 1, Case 2, and Case 3 were 0.8750, 0.8963, and 0.9111, respectively, indicating that intermittent operation resulted in higher flux than continuous operation. However, it should be noted that the effective operating time for Case 2 and Case 3 was 480 min due to the 120 min of the shutdown period. For a fair comparison, the flux at 480 min for Case 1 was compared, which was 0.8824, confirming that it was still lower than those for Case 2 and Case 3.

### 3.2. Foulant Layer Thickness after Intermittent Operation

To elucidate the reason for the increased flux in the intermittent operation, the thickness of the foulant layer was analyzed by OCT and the image analysis technique. After 4 h of continuous operation, the image of the foulant layer was obtained, as shown in Figure 4a, and the average thickness was calculated to be 10.4 μm. As can be seen in Figure 4b, the foulant layer thickness was found to be reduced to 8.42 μm when the operation was shut down for 120 min and then restarted (Case 2). A further reduction in the thickness (7.47 μm) was observed in Figure 4c when the membrane was stored in DI water after the shutdown and then restarted (Case 3). These results are consistent with the flux changes shown in Figure 3d.

### 3.3. Changes in Flux and Thickness in Continuous Operation

To further investigate the behaviors of flux and the foulant layer thickness, the RO system was continuously operated for 40 h and the OCT analysis was repeated. The same feed solution and the operating conditions were used here as in the previous experiments. A steady decrease in flux was observed over time, resulting in a flux reduction of approximately 24%, as shown in Figure 5a. The flux reduction was fast at the beginning and then slowed down after 4 h. A similar trend was observed for foulant layer thickness, as can be seen in Figure 5b. The thickness rapidly increased to 15.9 μm within 3 h and gradually reached 21.2 μm after 40 h.

The following model equation was used to derive a quantitative relationship between flux and the fouling layer thickness. According to the resistance-in-series model and the cake formation theory, the flux (*J*) can be described by:(1)$$J=\frac{\Delta P-\Delta \pi }{\eta \left(R_{m}+R_{c}\right)}=\frac{\Delta P-\Delta \pi }{\eta \left(R_{m}+\frac{\alpha_{c}m_{c}}{A_{m}}\right)}$$
where *ΔP* is the transmembrane pressure, *Δπ* is the osmotic pressure difference between the feed and permeate sides, η is the viscosity of the permeate, *R_m_* is the intrinsic resistance of the membrane, *R_c_* is the resistance of the foulant (cake) layer, $\alpha_{c}$ is the specific resistance of the foulant, *m_c_* is the mass of the foulant, and *A_m_* is the membrane area. The foulant layer thickness (τ) may be related to the *m_c_*:(2)$$\tau =\frac{m_{c}}{\rho_{c}A_{m}}$$
where *ρ*_c_ is the density of the foulant. Accordingly, *J_v_* is expressed as a function of τ:(3)$$J=\frac{\Delta P-\Delta \pi }{\eta \left(R_{m}+\alpha_{c}\rho_{c}\tau \right)}$$

Rearranging Equation (3), the following equation is obtained:(4)$$\frac{1}{J}=\frac{1}{\left(\frac{\Delta P-\Delta \pi }{\eta \rho_{c}\alpha_{c}}\right)}\left(\tau +\frac{R_{m}}{\alpha_{c}\rho_{c}}\right)=\left(\frac{\eta \rho_{c}}{\Delta P-\Delta \pi }\right)\left(\alpha_{c}\right)\tau +\left(\frac{\eta_{c}R_{m}}{\Delta P-\Delta \pi }\right)$$

Thus, the thickness is inversely proportional to the flux:(5)$$\tau \propto \frac{1}{J}$$

Figure 6 shows the correlation between the thickness (t) and the reciprocal value of the flux (1/*J*). A linear relationship was observed, as expected in Equation (4), confirming that the thickness measured by OCT is useful to analyze the flux changes. It should be noted that the data points in the late stage showed a different trend from those in the early stage, resulting in two regression lines with different slopes. This can be attributed to changes in the foulant layer characteristics with time. As the foulant layer compression occurred over time, the *α_c_* increased, leading to an increase in the slope of the regression line.

### 3.4. Changes in Flux and Thickness in Intermittent Operation

Figure 7 shows the evolution of the flux over time in the intermittent operation in comparison with the results of the continuous operation. The effective operating time of intermittent operation was set to 40 h, which implies that the period of downtime was excluded from this. There were four shutdowns and restarts because the operation was stopped every 8 h. In the intermittent operation, when the membrane was kept during the downtime using the influent water (Case 2), it was observed that the flux increased at each restart. This led to an increase in the final flux at 40 h of the effective operation time, as can be seen in Figure 7a. The increase in flux was even greater when the membrane was stored with DI water in intermittent operation (Case 3). This effect accumulated over repeated starts and stops, improving the final flux, as shown in Figure 7b.

The previous results indicate that intermittent operation causes flux recovery. To clarify the specific mechanism, the changes in fouling layer thickness before and after intermittent operation were tracked by OCT. As shown in Figure 8, the images of the fouling layer were obtained and analyzed in each condition. These images were used to calculate the thickness of the foulant layer, which are shown in Figure 9 as a function of time in each operation cycle. In Case 2, as shown in Figure 9a, the thickness increased over time during the operation period. When the operation was stopped and the membrane was stored in the feed water for 16 h, the thickness slightly increased. However, immediately after resuming operation, the foulant layer thickness decreased. These results indicate that the flux improvement in Case 2 is attributed to the reduction of fouling upon restart.

Figure 9b shows the profile of the foulant layer thickness in each of the operating cycles for Case 3. Similar to Case 2, the thickness increased with time during the operation period. However, unlike the previous case, a decrease in the thickness was observed when the operation was stopped and DI water was supplied. Little increase in thickness was observed even when the membrane was stored in DI water. A further decrease in thickness was observed when operation was resumed. These results imply that the flux improvement in Case 3 is due to the reduction of fouling in both the stop and restart periods.

Based on these findings, the mechanism for flux improvement in intermittent operation was proposed, as shown in Figure 10. In Case 2, where the membrane was stored in the feed water during the downtime, the foulant removal occurred only when the operation was restarted. Additional foulant deposition appears to occur during the storage period because the membrane was in contact with the foulant-containing feed water. On the other hand, foulant removal occurred during the shutdown and restart periods in Case 3, where the membrane was stored in DI water during the shutdown. Replacing the feed water with DI water caused the foulants to be removed from the membrane. During the storage period, there was no additional foulant deposition due to the use of the DI water.

More quantitative data on the change of the foulant layer thickness are shown in Figure 11, where the reduction of the foulant layer thickness was calculated and compared in each operating cycle. In Case 2 (Figure 11a), the reduction in thickness was observed in the restart period, while the increase (or negative reduction) in thickness was obtained in the storage period. The net thickness reduction is the difference between these two values. On the other hand, in Case 3 (Figure 11b), the reduction in thickness was observed in both the storage and restart periods, and the net reduction is the sum of these two values. Correspondingly, the reduction in the foulant layer thickness was greater in Case 3 than in Case 2.

Figure 12 shows the relationships between the foulant layer thickness (*τ*) and the reciprocal value of the flux (1/J) for Case 2 and Case 3. A linear relationship between the two variables was found in each case, albeit with deviations. In Figure 6, the slope of the regression line was 0.0255, while those for Case 2 and Case 3 were 0.0156 and 0.0183, respectively. As shown in Equation (4), the slope of the regression lines is proportional to *α*_c_, indicating that the foulant layer in the continuous operation had a greater *α*_c_ value than those in the intermittent operations. It can be interpreted that the membrane fouling layer became somewhat loose due to the shutdown and restart periods in the intermittent operation.

### 3.5. 3D OCT Images with AFM Results

Figure 13a shows the 3D OCT images of the foulant layer in the continuous operation (Case 1). As described earlier, the dimensions of the image were 1946 μm × 114 μm. The surface of the foulant layer showed a large roughness and a tendency to become denser towards the bottom. The 3D OCT images in the intermittent operations (Case 2 and Case 3) are shown in Figure 13c,e. The foulant layer in Case 1 appeared to be the thickest, while that in Case 3 seemed to be the thinnest. The foulant layer in Case 2 showed an intermediate thickness. These results are in accordance with the 2D OCT results in Figure 8.

The AFM analysis was performed simultaneously on the same membranes, and the results are shown in Figure 13b,d,f. Note that the AFM images are smaller than the 3D OCT images, so a direct comparison is not possible. The roughness values for these membranes were found to be 0.754 μm (Case 1), 0.806 μm (Case 2), and 1.065 μm (Case 3), indicating that the foulant layer in the continuous operation was smoother than those in the intermittent operation. This is attributed to the removal of the foulant from the membrane surface during the intermittent operation. Since the foulant detachment occurred in an uneven manner, it is likely that the surface of the foulant resulted in a higher roughness. In addition, the difference in the morphological properties between the continuous and intermittent operation can be related to the α_c_ for the foulant layers. It is also likely that information on the morphology by OCT may be used to identify the type of fouling. Of course, further studies are required to explore these possibilities.

## 4. Conclusions

In this paper, RO fouling was investigated under continuous and intermittent operating conditions using an OCT-based approach. The flux was found to be improved after shutdown and restart in intermittent operation compared to continuous operation. The intermittent operation also showed a decrease in the thickness of the fouling layer. A linear relationship between the thickness (*τ*) and the reciprocal value of the flux (1/*J*) was obtained, indicating that the thickness measured by OCT is useful for analyzing the flux variations. The intermittent operation with repeated stops and restarts also showed an increase in flux and a decrease in the fouling layer thickness. When the membrane was kept in the feed solution during the shutdown period, the reduction of the fouling layer was observed only during the restart period. On the other hand, when the membrane was stored in DI water during the shutdown period, the reduction of the fouling layer was observed during both the shutdown period and the restart period. This resulted in a higher final flux in the latter case than in the former. Based on these results, a mechanism for reducing fouling in intermittent operation was proposed, and regression equations showing the relationship between flux and fouling layer thickness were obtained. Meanwhile, 3D OCT images and AFM results provided three-dimensional information and morphological characteristics of the fouling layer.

## Figures and Tables

**Figure 1 membranes-13-00392-f001:**
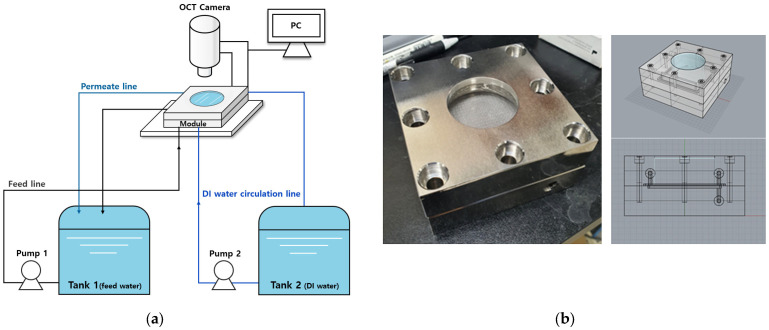
(**a**) Schematic diagrams of RO test apparatus. (**b**) Membrane module for OCT analysis.

**Figure 2 membranes-13-00392-f002:**
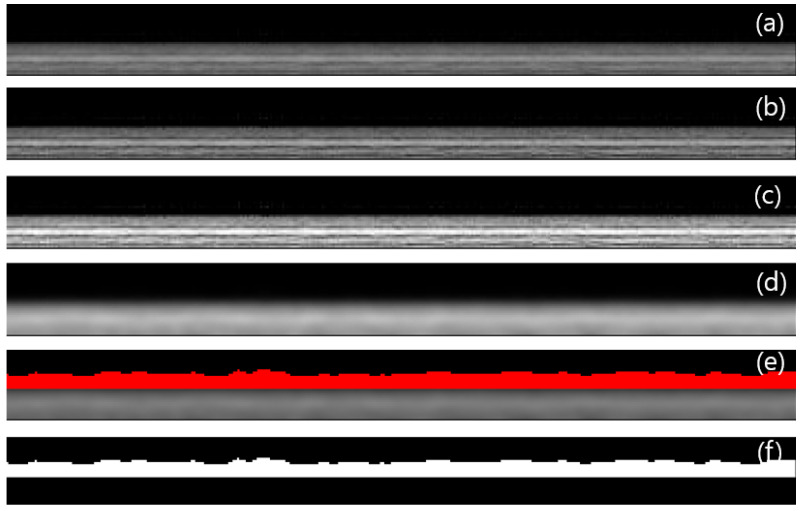
Example of image analysis for foulant layers on the membrane: (**a**) raw OCT image, (**b**) after Unsharp mask, (**c**) after contrast-limited adaptive histogram equalization (CLAHE), (**d**) after Gaussian blur, (**e**) thresholding image (red color is the foulant layer), and (**f**) fouling image (white color is the foulant layer).

**Figure 3 membranes-13-00392-f003:**
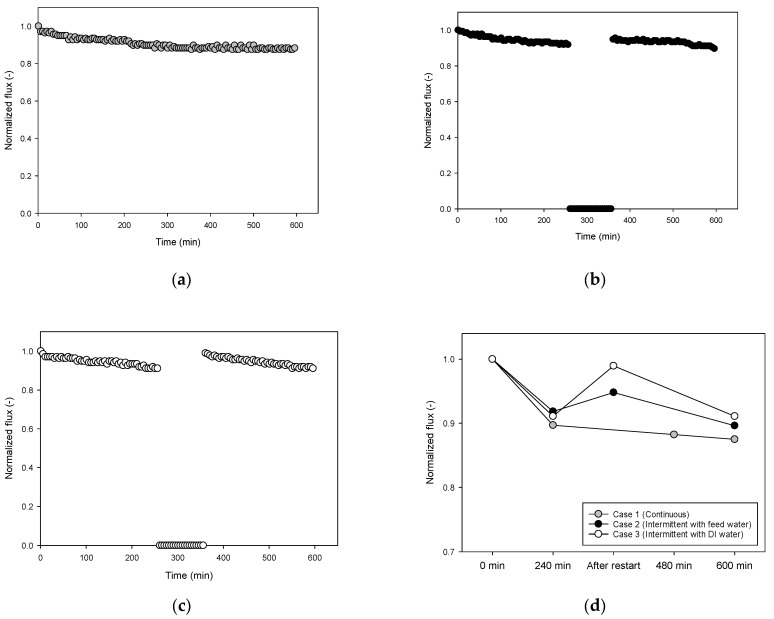
Time-dependent flux profiles for 600 min of RO operation. (**a**) Continuous operation (Case 1), (**b**) intermittent operation with feed water storage (Case 2), (**c**) intermittent operation with DI water storage (Case 3), and (**d**) comparison of flux variations.

**Figure 4 membranes-13-00392-f004:**
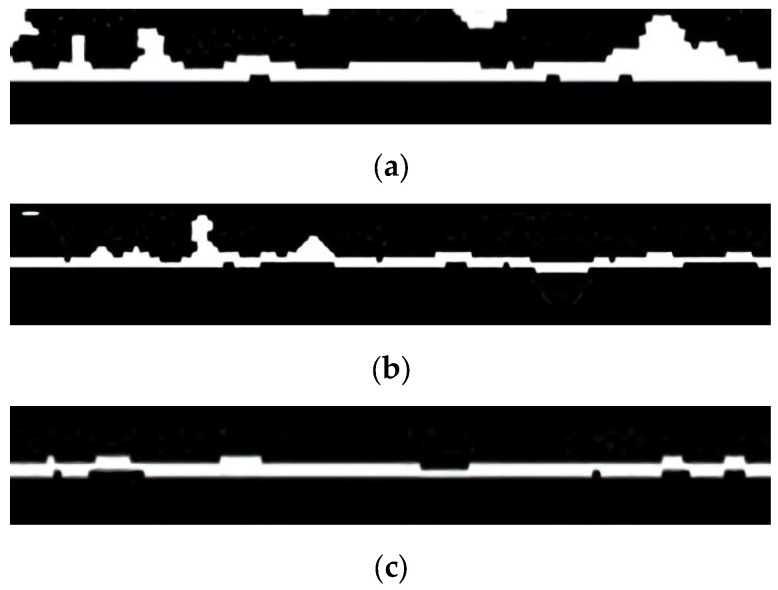
Images of the foulant layer by OCT (after thresholding). The white areas in the images represent the foulant layers. (**a**) Continuous operation for 4 h. (**b**) Restart after the membrane storage in the feed water (Case 2). (**c**) Restart after the membrane storage in DI water (Case 3).

**Figure 5 membranes-13-00392-f005:**
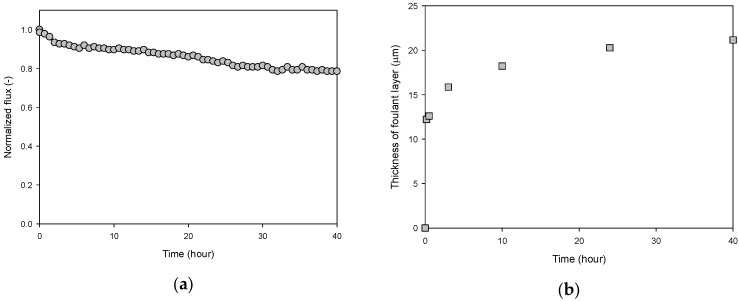
Time-dependent profiles for flux and foulant layer thickness for 40 h of continuous RO operation: (**a**) flux and (**b**) foulant layer thickness.

**Figure 6 membranes-13-00392-f006:**
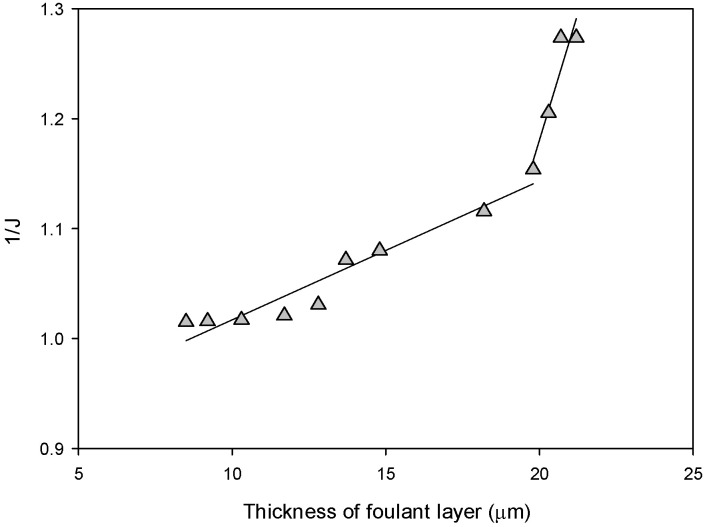
Correlation between the foulant layer thickness and the reciprocal of flux for continuous RO operation.

**Figure 7 membranes-13-00392-f007:**
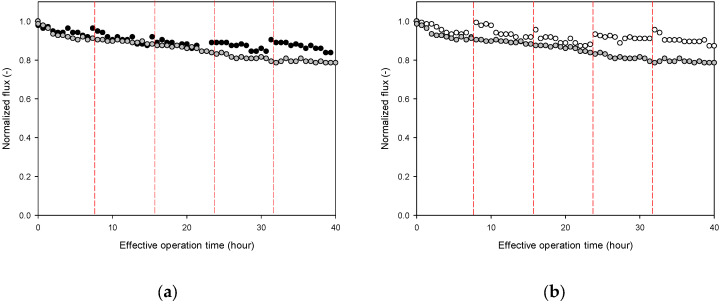
Time-dependent profiles for flux and foulant layer thickness for 40 h of intermittent operation. The red dashed line represents the time for stopping and restarting. black color is Case 2 operation, gray color is continuous operation, and white color is Case 3 operation. (**a**) RO operation with membrane storage in the feed water (Case 2). (**b**) RO operation with the membrane storage in DI water (Case 3).

**Figure 8 membranes-13-00392-f008:**
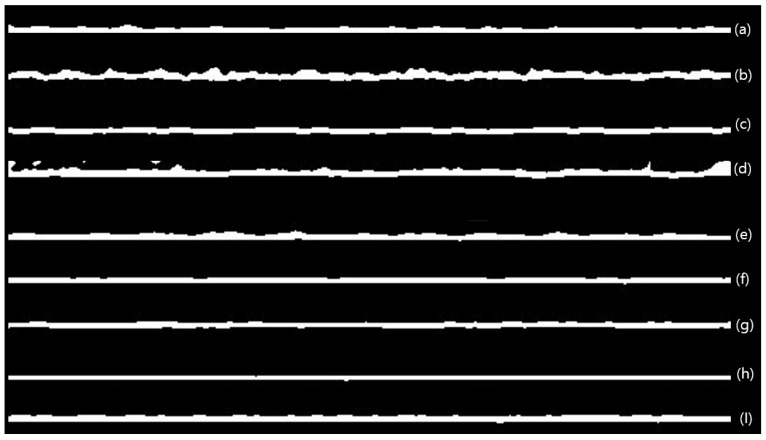
Images of the foulant layer by OCT (after thresholding). The white areas in the images represent the foulant layers. (**a**) Case 2—After 8 h (before shutdown). (**b**) Case 2—After shutdown, 16 h. (**c**) Case 2—Restart. (**d**) Case 2—After a 40 h operation time. (**e**) Case 3—After 8 h (before shutdown). (**f**) Case 3—After DI cleaning, 5 min. (**g**) Case 3—After shutdown, 16 h. (**h**) Case 3—Restart. (**i**) Case 3—After a 40 h operation time.

**Figure 9 membranes-13-00392-f009:**
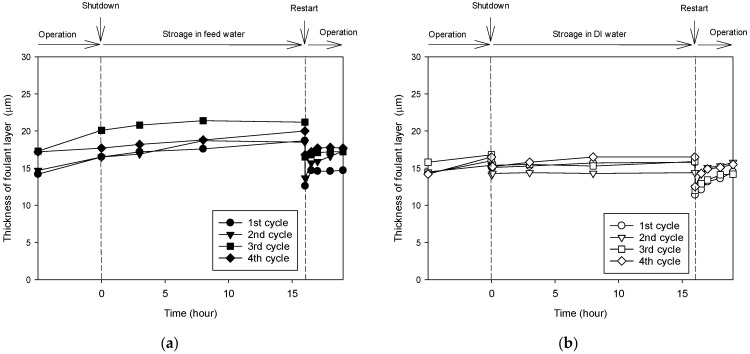
Changes in the foulant layer thickness in the intermittent operating cycle. (**a**) RO operation with membrane storage in the feed water (Case 2). (**b**) RO operation with the membrane storage in DI water (Case 3).

**Figure 10 membranes-13-00392-f010:**
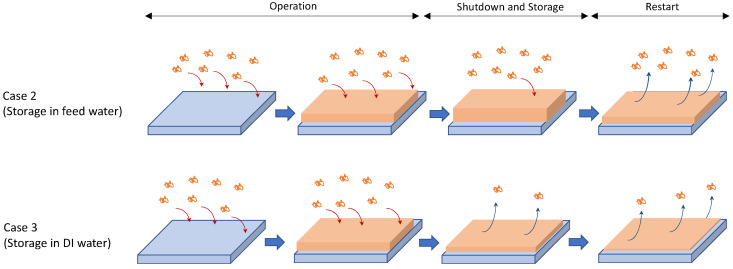
Proposed mechanism for flux enhancement and foulant removal by intermittent operation.

**Figure 11 membranes-13-00392-f011:**
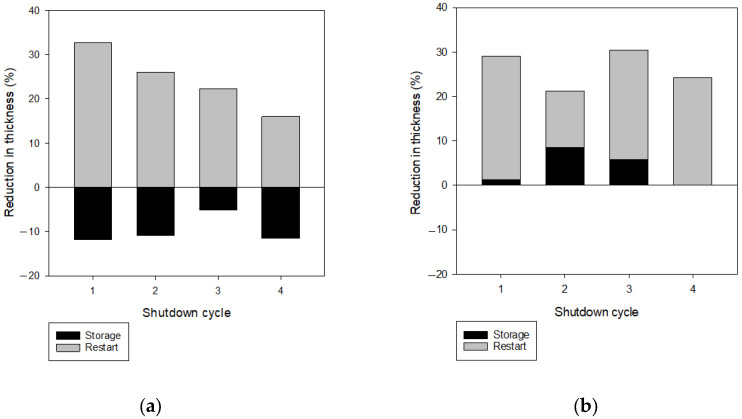
Analysis of the reduction in foulant layer thickness during intermittent operation. (**a**) RO operation with membrane storage in the feed water (Case 2). (**b**) RO operation with the membrane storage in DI water (Case 3).

**Figure 12 membranes-13-00392-f012:**
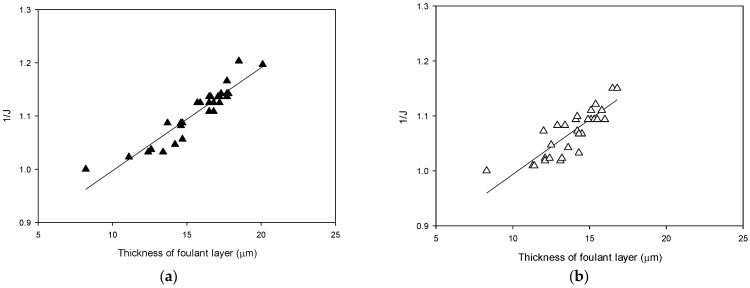
Correlation between the foulant layer thickness and the reciprocal of flux for intermittent RO operation. (**a**) RO operation with membrane storage in the feed water (Case 2). (**b**) RO operation with the membrane storage in DI water (Case 3).

**Figure 13 membranes-13-00392-f013:**
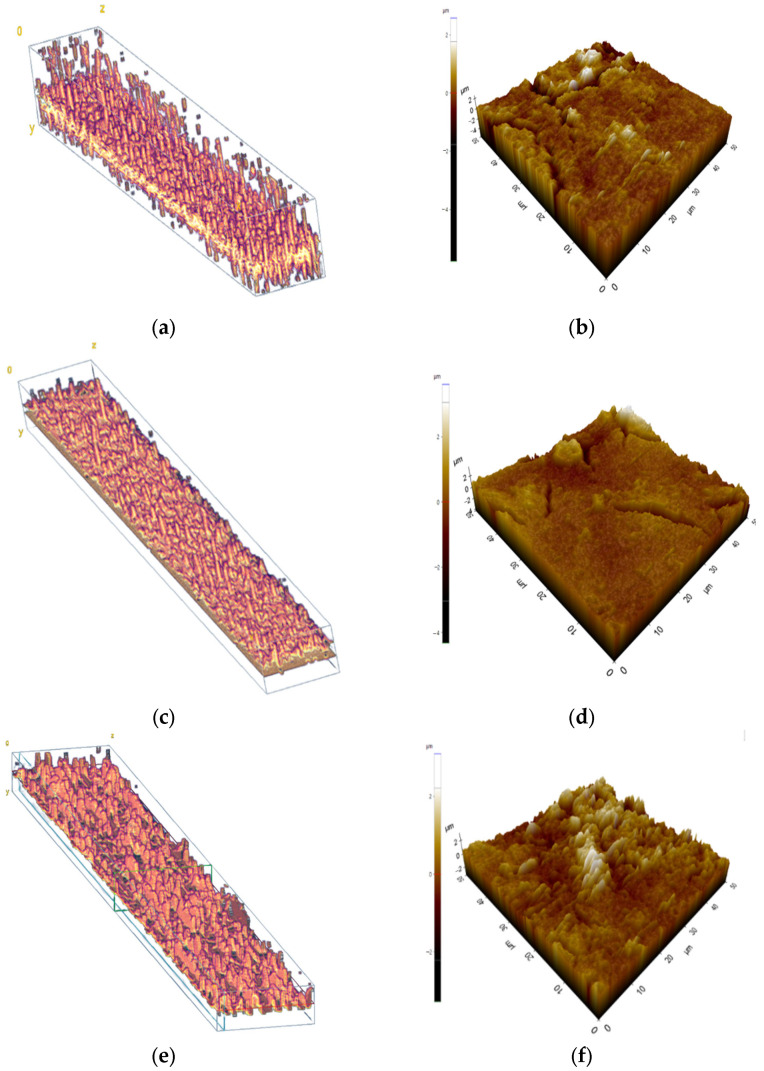
In OCT images (**a**,**c**,**e**), the yellow color represents the membrane, while the pink color above the yellow corresponds to the foulant layer on top of the membrane. In AFM images (**b**,**d**,**f**), brighter colors indicate that the foulant layers are stacked higher on the membrane compared to other areas (**a**) 3D OCT image for continuous RO operation (Case 1), (**b**) AFM image for continuous RO operation (Case 1), (**c**) 3D OCT image for RO operation with membrane storage in the feed water (Case 2), (**d**) AFM image for RO operation with membrane storage in the feed water (Case 2), (**e**) 3D OCT image for RO operation with the membrane storage in DI water (Case 3), and (**f**) AFM image for RO operation with the membrane storage in DI water (Case 3).

**Table 1 membranes-13-00392-t001:** Summary of operating conditions for RO experiments.

Total Duration	Time in Each Cycle	Case 1 (Continuous)	Case 2 (Intermittent Operation with Feed Water Storage)	Case 3 (Intermittent Operation with DI Water Storage)
6 h	4 h	RO in operation	RO in operation	
	2 h		RO stored in feed water	RO stored in DI water
40 h	8 h	RO in operation	RO in operation	
	16 h		RO stored in feed water	RO stored in DI water
	8 h		RO in operation	
	16 h		RO stored in feed water	RO stored in DI water

## Data Availability

Not applicable.

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
