# Peer review of "Application of Optical Coherence Tomography (OCT) to Analyze Membrane Fouling under Intermittent Operation"

_membranes, 2023, doi:10.3390/membranes13040392_

Round 1
Reviewer 1 Report
In this paper, the authors used the OCT technique to analyze the membrane fouling phenomenon for intermittent operation. The subject of more scientific observation of the existing membrane fouling phenomenon is very interesting and fresh.
Author Response
We appreciate you about the reviewers for your precious time in reviewing our paper and providing valuable comments.
Reviewer 2 Report
The manuscript titled “Application of Optical Coherence Tomography (OCT) to analyze membrane fouling under intermittent operation” and written by Song Lee et al. is interesting. In general, the paper is well written and structured, however, it has some shortcomings that should be addressed by the authors. I recommend a moderate revision based on the following comments:
1. In page 1, line 35, reverse osmosis was abbreviated to RO however, in the same page, line 40 it is written reverse osmosis. Please, use the abbreviations properly, revise the entire document.
2. In page 2, second paragraph the authors mentioned that one of the main challenges of RE-powered RO systems is the intermittency of renewable energy. They should also mention that not only the intermittency but also the variability of the energy could me the RO system to work under variable operating conditions (Desalination, 532, 115715; Computers & Chemical Engineering 153, 107441) which makes more difficult to estimate the performance in terms of permeate quality and production.
3. In page 2, lines 50-56 the authors wrote that there are not much information about fouling in RO systems working under intermittent operation. This is true, fouling caused performance decrease in RO systems which produce permeate flow decrease or feed pressure increase to keep the permeate production constant. There are relevant papers related with intermittent operation that the authors have not mentioned (Journal of Membrane Science, 636, 119556; Desalination 489, 114526; Renewable Energy, 135, pp. 108–121; Desalination, 435, pp. 188–197). These studies should be included and commented.
4. The authors used a little piece of membrane. Would be a problem for the OCT technology to evaluate fouling in real time using industrial spiral wound membrane modules inside pressure vessels?
5. Why the authors used 16 g/L of NaCl? This is a quite high concentration considering BWRO system.
6. Page 4, line 144, please, give space between 22 and sec. Abbreviation of hour is h and of second is s. Please revise the entire manuscript.
7. The OCT technology allowed to see the thickness increment of foulant layer during the operation. Does this have any application in full-scale RO systems? Usually the real-time indicator of fouling is the pressure drop in the stages od the RO system or the permeate flux decline. This does not give any information about the type of fouling (organic, inorganic, biofouling..). Would be possible to identify the type of fouling in RO systems in real time with the OCT technology?
